# Targeted Mutational Analysis of Circulating Tumor DNA to Decipher Temporal Heterogeneity of High-Grade Serous Ovarian Cancer

**DOI:** 10.3390/cancers14153697

**Published:** 2022-07-29

**Authors:** Lara Paracchini, Laura Mannarino, Luca Beltrame, Fabio Landoni, Robert Fruscio, Tommaso Grassi, Maria Luisa Dalessandro, Maurizio D’Incalci, Sergio Marchini

**Affiliations:** 1Department of Biomedical Sciences, Humanitas University, Via Rita Levi Montalcini 4, Pieve Emanuele, 20072 Milan, Italy; lara.paracchini@humanitasresearch.it (L.P.); laura.mannarino@humanitasresearch.it (L.M.); 2Laboratory of Cancer Pharmacology, IRCCS Humanitas Research Hospital, Via Manzoni 56, Rozzano, 20089 Milan, Italy; luca.beltrame@humanitasresearch.it (L.B.); maria.dalessandro@humanitasresearch.it (M.L.D.); sergio.marchini@humanitasresearch.it (S.M.); 3Department of Obstetrics and Gynaecology, Università degli Studi Milano-Bicocca, San Gerardo Hospital, 20900 Monza, Italy; fabio.landoni@unimib.it (F.L.); robert.fruscio@unimib.it (R.F.); grassi-400807@asst-monza.it (T.G.)

**Keywords:** high-grade serous ovarian cancer, liquid biopsy, targeted resequencing

## Abstract

**Simple Summary:**

The issue of spatial and temporal heterogeneity of high-grade serous ovarian cancer (HGS-EOC) has hampered the possibility to shape the molecular portrait of relapsed disease, which ultimately impacts our ability to develop a more rational second-line treatment. Liquid biopsy offers the unique opportunity to track tumor evolution over time and infer the dynamic changes of tumor clonal architecture. Differently from other tumors, no actionable driving lesions characterize HGS-EOC, thus genome-scale analysis like whole-exome sequencing is not compatible with the clinical turnaround time. In the present work, we provided a novel framework based on the analysis of both qualitative and quantitative features of circulating tumor DNA (ctDNA) in order to identify, at the time of molecular relapse, the early genetic vulnerabilities that will characterize the clinical recurrence and thus be amenable of a more rational second-line treatment.

**Abstract:**

We have previously demonstrated that longitudinal untargeted analysis of plasma samples withdrawn from patients with high-grade serous ovarian cancer (HGS-EOC) can intercept the presence of molecular recurrence (TRm) earlier than the diagnosis of clinical recurrence (TRc). This finding opens a clinical important temporal window to acquire through plasma sample analysis a real-time picture of those emerging molecular lesions that will drive and sustain the growth of relapsed disease and ultimately will confer resistance. In this proof of principle study, the same genomic libraries obtained at the diagnosis (T0), TRm and TRc were further analyzed by targeted resequencing approach to sequence the coding region of a panel of 65 genes to provide longitudinal analysis of clonal evolution as a novel strategy to support clinical decisions for the second-line treatment. Experiments were performed on plasma and tumor tissues withdrawn on a selection of previously analyzed cohorts of cases (i.e., 33 matched primary and synchronous lesions and 43 plasma samples from 18 patients). At T0, the median concordance of mutations shared by each tumor tissue biopsy and its matched plasma sample was 2.27%. This finding confirms the limit of a single tumor biopsy to be representative of the entire disease, while plasma analysis can recapitulate most of the main molecular lesions of the disease. A comparable scenario was observed during longitudinal analysis, where, with the exception of the *TP53* gene and germline mutations in *BRCA1/2* genes, no other gene shared the same locus specific gene mutation across T0, TRm and TRc time points. This high level of temporal heterogeneity has important implications for planning second-line treatment. For example, in three out of 13 cases, plasma ctDNA analysis at TRm or TRc reported acquired novel variants in the *TP53BP1* gene not present at T0. In particular, patient 21564, potentially eligible for PARP-inhibitor (PARPi) treatment at the time of diagnosis (*BRCA1* c.5182delA mutation), would unlikely respond to these drugs in second-line therapy due to the presence of eight distinct *TP53BP1* variants in plasma samples collected TRc. This study demonstrates that liquid biopsy provides a real-time molecular picture to intercept those actionable genetic vulnerabilities or drug resistance mechanisms that could be used to plan a more rational second-line treatment.

## 1. Introduction

High-grade serous epithelial ovarian cancer (HGS-EOC), the most common and lethal subtype of ovarian cancer, is a systemic disease often diagnosed when multiple metastatic lesions (i.e., synchronous lesions) have already disseminated in the abdominal cavity (stage III–IV). Although approximately 80% of cases are initially responsive to platinum (Pt)-based chemotherapy, the majority of them relapse with a progressive Pt-resistant disease. One of the most challenging issues that hampers the possibility to effectively treat the relapsed disease is the lack of biological information about tumor recurrence. To date, the Pt-free interval (PFI), an empirical measure of the time lagging between the end of front-line chemotherapy and relapse, is the only and widely accepted parameter to predict patient outcome and sensitivity to Pt second-line chemotherapy [1].

Longitudinal acquisition of multiple tissue biopsies is clinically impracticable, and this limits the possibility to develop second-line treatments based on the knowledge of acquired tumor molecular architecture.

The use of liquid biopsy has emerged as an attractive non-invasive tool able to overcome this issue, allowing to monitor tumor evolution over time. Particularly, the analysis of circulating tumor DNA (ctDNA) isolated from plasma in the pool of circulating free DNA (cfDNA) has been proven to be a successful approach in anticipating recurrence and in predicting therapy response [2,3,4]. ctDNA, defined as the fraction of tumor-released DNA fragments, is shed by the tumor cells as a result of apoptosis, necrosis and active secretion processes [5] and can be detected and quantified in cfDNA through the research of genetic alterations that are specific for cancer tumor cells [6]. The analysis of ctDNA has been extensively studied in cancer patients as it represents an alternative to tissue biopsies for the analysis of the dynamic changes that occur in the genome between primary tumor and/or relapsed disease [7,8,9,10,11,12]. Moreover, as ctDNA can be obtained through a non-invasive procedure, the longitudinal plasma collections during treatment of follow-up allow a real-time monitoring of drug response and risk of relapse in several cancer types, making it an ideal biomarker [9,10]. Differently from other solid tumors, the development of liquid biopsy in the clinical setting of HGS-EOC has been limited till now [13,14].

The majority of the HGS-EOC genome is characterized by marked chromosomal instability and, with the exception of the *TP53* gene, no other clonal driver mutations are known [15]. These features have shifted the focus of liquid biopsy towards developing an untargeted approach versus a targeted one. Recently, we have demonstrated that the low-pass whole-genome sequencing (sWGS) approach is a suitable tool to intercept the early molecular traces of the relapsing disease in the plasma of patients with a diagnosis of HGS-EOC [2]. The calculation of tumor fraction (TF) allowed us to anticipate the diagnosis of the relapsed disease by almost 240 days (range: 39–491) compared to conventional biochemical (CA-125) and radiological (transvaginal sonography) tools [2]. Unfortunately, TF provides only quantitative information of the amount of ctDNA with no qualitative information on the genomic features of the emerging resistant clones, i.e., presence of novel actionable vulnerabilities or somatic variants that could confer resistance/sensitivity to frontline therapy.

In this proof of concept study, we have developed a two-step workflow in which targeted resequencing libraries generated previously from sWGS were interrogated at the time of molecular/clinical recurrence (TRm and TRc, respectively) to get quantitative information of the genomic feature of relapsed disease. This workflow has the advantage to derive quantitative and qualitative genomic information from a single plasma withdrawal, with a considerable cost-benefit advantage.

## 2. Materials and Methods

### 2.1. Patient Selection

A longitudinal retrospective cohort of 18 HGS-EOC patients derived from a larger group of 46 patients previously analyzed [2] was selected for this study (Appendix A). The CONSORT diagram depicted in Appendix A describes the process of case selection. Particularly, all included patients received a diagnosis of advanced stage (stage III/IV) HGS-EOC according to the International Federation of Gynecological and Obstetrics (FIGO) classification guidelines [16] and underwent debulking surgery at the Obstetrics and Gynecology Department of San Gerardo Hospital (Monza, Italy). After the local scientific ethical committee approval and the obtainment of written informed consent from all enrolled patients, a total number of 33 biopsies derived from both primary tumor masses and synchronous lesions and 43 plasma samples were included for the purposes of this study. The study has been carried out following the Declaration of Helsinki.

### 2.2. Tumor DNA and cfDNA Library Preparation

Targeted re-sequencing analysis was performed on sWGS libraries generated from DNA purified from both single-spot tumor biopsies and matched plasma samples, as previously described [2]. Briefly, 1000 ng derived from twelve barcoded sWGS libraries (approximately 80 ng per sample) were pooled together and hybridized for 16 h with custom probes covering the entire coding regions of 65 preselected genes. The complete list of selected genes is reported in Appendix A. Capture probes were designed using a free tool (Hyper Design, Roche. Basel, Switzerland) and optimized in order to have 100% coverage in the regions of interest (SeqCap EZ, Roche. Basel, Switzerland). Following the manufacturer’s instructions, final libraries were generated, their amount assessed using a fluorescent dye (HS dsDNA Qubit, Life Technologies, Carlsbad, CA, USA), and library size distribution evaluated by exploiting Tape Station 4200 (Agilent Technologies, Santa Clara, CA, USA). Libraries were then sequenced on a sequencing platform (NextSeq500, Illumina, San Diego, CA, USA) to reach a mean of coverage of 7000× for each sample.

### 2.3. Sequencing Data Analysis

Raw reads were aligned to the reference genome (hg38) with BWA [17]. A size selection filter was applied on aligned reads to only include cfDNA and exclude potential artifacts [2]. Variant calling was performed with MuTect2 [18] and VarDict [19]. Then, results from both callers were merged into a single unified call set. When analyzing plasma samples, variant calling parameters were adjusted to improve sensitivity, in particular, variant filtering from VarDict [19] was adjusted to include very low (<1%) fraction variants. Called variants were annotated with the Variant Effect Predictor (VEP) [20] and loaded into a custom in-house database for further processing.

### 2.4. Variant Filtering and Prioritization

To remove sequencing and library bias artifacts, an aggressive filtering strategy was employed. Firstly, sample-specific variant allele fraction (VAF) thresholds were defined as the known tumor fraction (TF) (as published in Paracchini et al. [2]) increased by 5% to account for sequencing errors. Secondly, only variants with at least five supporting reads were selected, and likely germline variants were removed, as previously described [21], and those present in the general population (>1%) were also excluded.

To exclude library artifacts, which appear as apparent insertions and deletions, we removed all variants which satisfied these criteria:deletion of one base upstream of a repeated sequence of at least 3 of the same nucleotides;insertion of one or two bases upstream of a repeated sequence of at least 3 of the same nucleotides;single-nucleotide variant with flanking repeated sequences of at least 3 nucleotides;inframe insertion/deletions of repeats occurring at least twice upstream or downstream to the variant locus;frameshifts affecting more than 60% of the sample cohort.

To identify putative pathogenic variants, the data were uploaded to the Cancer Genome Interpreter [22] to determine their tumor driver status. CGI results were filtered to only include known driver variants, variants with high confidence (“tier 1”), and variants with medium confidence but supporting evidence (“tier 2”). Oncoprints were drawn using CoMut [23].

### 2.5. Jaccard Similarity Score

In order to define the concordance of *variant calls* between pairs of plasma sample in each patient, we calculate the *Jaccard score*, defined as follows:$$Jaccard\;score_{a,b}=\frac{variant\;calls_{a}⋂variant\;calls_{b}}{variant\;calls_{a}⋃variant\;calls_{b}}$$
considering as overlapping variants those with the same locus, base change and annotation.

### 2.6. Lolliplots

Lolliplots were done with the trackViewer R package [24]. Protein domains were retrieved from the protein database of the EMBL-EBI Pfam [25]

## 3. Results

### 3.1. Cohort Description

A retrospective cohort of 18 patients with a histologically confirmed diagnosis of HGS-EOC was selected for this study. The main clinical, pathological, and demographic information of patients enrolled are reported in Table 1 and were as previously described [2]. Briefly, the entire cohort is composed of 33 snap-frozen tumor biopsies—with a mean of two synchronous biopsies per patient—collected during the debulking (61%) or interval surgery (39%) and 43 plasma samples collected at the time of diagnosis (T0) and during patients’ follow-up.

The study design is schematically reported in Figure 1. Briefly, the entire cohort was divided into two subsets, named from now onwards as cohort A and cohort B, to achieve the two following aims: (i) to quantify at T0 the spatial heterogeneity between ctDNA and tumor DNA (cohort A); (ii) to evaluate ctDNA temporal heterogeneity and the extension to which it impacts on second-line therapeutic options (cohort B).

As detailed in Appendix A, cohort A is composed of 12 patients naive to chemotherapy for whom at least one primary and synchronous tumor lesion with matched plasma samples were available. Cohort B includes only plasma samples withdrawn from the 18 patients enrolled in the study. For each patient, we have at least two plasma samples longitudinally collected during patients’ follow-up. Particularly, for cohort B plasma samples were collected: (*i*) at the time of diagnosis or at the first time available during patients’ follow-up (*n* = 3, Appendix A); (*ii*) at the time of TRm (defined as 20% of increase in the percentage of TF compared to the previous point) [2] and/or (*iii*) at time of TRc.

### 3.2. Mutational Landscape of Matched Tissues and Plasma Samples in HGS-EOC Patients

Targeted resequencing analysis was performed on an arbitrarily selected panel of 65 genes representative of the main pathways reported to be deregulated in HGS-EOC, some of them being potentially amenable to therapeutic intervention (Appendix A). SNV profiles generated at diagnosis from cohort A were investigated to define the extent to which ctDNA recapitulates the complex and heterogeneous genomic profile of the disease. To this aim, we compared the mutational profiles of all nonsynonymous variants called in each tissue sample at T0 with those in the matched plasma sample. Data reported in Figure 2 show that, except for clonal pathogenic variants of the *TP53* gene, plasma samples exhibited a clearly different mutational landscape compared to their matched tissue biopsies. In line with the hypothesis that ctDNA recapitulates the genomic profile of a disease characterized by multiple systemic lesions present at once, plasma samples carried significantly more variants than each matched tumor biopsy (median of 41.5 and 7 variants per sample, respectively; Mann–Whitney test *p* < 0.001). This difference prompted us to investigate the extent to which plasma analysis can recapitulate the main genomic features of the single matched tumor lesion in detail. We divided somatic variants into three classes: (*i*) private, which refers to variants present only in a single sample (plasma or tissue); (*ii*) shared, which refers to those variants that are present in plasma samples and in at least one matched solid tumor biopsy; (*iii*) common, which refers to those somatic variants that are present in the plasma and in at least two matched biopsies. Globally, results suggest that the vast majority of SNVs (85.4%) were private to each single sample while the median concordance of shared mutations was 2.27%. Common variants represented only 0.1%. We next questioned whether the analysis of SNV profiles in plasma captures the marked spatial heterogeneity of HGS-EOCs better than the analysis of SCNAs [2]. Appendix A shows that plasma SCNA profiles have a percentage of concordance among biopsies and matched plasma sample higher than SNVs profiles, thus confirming the original finding that TF is a bona fide representative snapshot of the molecular portrait of HGS-EOC [2].

### 3.3. Longitudinal Plasma Analysis to Overcome the Issue of Temporal Heterogeneity

Focusing on targeted resequencing data generated within cohort B, we investigated the possibility to intercept in plasma samples those genomic vulnerabilities that characterize the molecular portrait of the relapsed disease. This is the second aim of the work that meets an important clinical need. We initially questioned whether the overall amount of SNVs per patient, generally known as tumor molecular burden (TMB), parallels TF changes [2] and thus could be associated with disease progression. The longitudinal distribution of TMB values (Appendix A) resulted as largely heterogeneous across samples and within each patient (median TMB per sample = 35, range 1–1158), with no correlation with the TF profile (Pearson’s correlation = −0.06, *p* > 0.5). This finding suggests that analysis of SNV in HGS-EOC is not as informative as TF to intercept the timing of TRm. In support of this notion, the nature of the identified SNV also showed the high heterogeneity of HGS-EOC. Indeed, we focused our attention on those SNVs that were shared among different time points of the same patient. The *Jaccard score* was used as a metric to estimate the percentage of co-occurring variants (ranging from 0 = no concordance, to 1 = full concordance). Figure 3a graphically depicts the *Jaccard score* of two representative samples, namely, 21556 and 21557, while data for the entire cohort are reported in Appendix A. Globally, the Jaccard similarity scores were low (max = 0.16), confirming the original hypothesis that temporal heterogeneity of SNV is the hallmark of HGS-EOC. Oncoprints reported in Figure 3b depict the pathogenic variants called for these two cases at T0, TRm and TRc (the complete list of pathogenic variants is available in Appendix A). In both cases, with the exception of clonal pathogenic variants in the *TP53* gene (c.843C > A and c.1045G > T, for 21556 and 21557, respectively) and a germline mutation in the *BRCA1* gene (c.4327C > T for 21556), no other pathogenic SNV was shared between T0 and TRm or TRc. These findings are in line with the hypothesis that *TP53* is the only driver pathogenic lesion of HGS-EOC and that the molecular picture of the relapsed disease is completely independent on the molecular portrait taken at diagnosis.

### 3.4. Dissecting Temporal Heterogeneity for Therapeutic Opportunities

As HGS-EOC is characterized by a marked temporal heterogeneity [26], liquid biopsy offers the unique opportunity to acquire real-time molecular portraits of those emerging genetic vulnerabilities that could be used to plan second-line therapy. Thus, selected pathogenic SNVs called in plasma samples at both TRm and TRc were further matched across different databases from official guidelines, clinical trials, case reports, or preclinical studies (see Section 2) for current knowledge on their potential actionability.

The lolliplots reported in Figure 4 refer to cases 21556 and 21557 as two paradigmatic examples. Focusing on sample 21556, of the 18 pathogenic variants called at T0 (Figure 3), only 6 were classified as actionable and, in particular, those affecting the *BRCA1/2, ATR,* and *POLE* genes (Figure 4a). All of them have been classified as tier 1 except for the R1371* variant in *POLE* that was classified as tier 2 (see Section 2). Variants in *BRCA1/2* and *POLE* have been associated with responsiveness to treatment with PARP-inhibitors (PARPi), Pt-based drugs and PD1 inhibitors, while both Y1419H and D1915Y variants in *ATM* were related to response to olaparib and temozolomide (Appendix A). The entire set of actionable alterations for each patient are reported in Appendix A.

Figure 4b depicts a completely different scenario for patient 21557. At TRm and TRc, the distribution of somatic variants (all classified as tier 1) in selected genes is more heterogeneous than that reported at T0. Focusing on pathogenic actionable mutations, the plasma sample at T0 carried actionable variants in *POLE* and *MSH3* genes, respectively, the first associated with responsiveness to PD1 inhibitors and pembrolizumab, while the latter to DNA-PKc inhibitors (Appendix A). At TRm, the plasma sample harbors one variant in the *ATM* gene, reported to be associated with response to Pt, temozolomide, PARPi, ATR-, and DNA-PKc inhibitors (Appendix A). Plasma analysis of TRc (that occurred 531 days later) called variants in the *PALB2* gene (D1141V) known to be involved in the response to PARPi, Pt and mitomycin, and a variant in the *FANCA* gene (S1264P) reported to be associated with responsiveness to olaparib (Appendix A).

Within this frame, we next reasoned how to exploit ctDNA genomic information to predict the molecular mechanisms of acquired drug resistance, in particular against PARPi, which are routinely used in the front-line therapy of HGS-EOC. Although this knowledge is largely incomplete, the concurrent presence of *BRCA1* mutation along with T*P53BP1* loss is at the moment one of the best characterized molecular mechanisms [27]. Analysis of cases enrolled in cohort A (Figure 2) shows that patient 21557 was the only one with a SNV called in the *TP53BP1* gene in all tumor biopsies but at T0. This analysis reveals once more the importance of plasma sample analysis at the baseline to dissect the complex spatial heterogeneity of HGS-EOC. Analysis of longitudinal cases enrolled in cohort B (Figure 5) revealed that, in 13 out of 18 samples, a pathogenic variant in the *TP53BP1* gene was observed but with a different time frame that can be summarized in the scenarios depicted in Figure 5.

In eight out of 13 cases, we observed that the *TP53BP1* gene is called as mutated at T0 (or T1) but resulted *wt* at TRm and TRc. To note, cases 21520 and 21564 acquired novel variants at TRm and TRc compared to those called at T0/T1. As a general comment, findings derived from the dynamic changes in the *TP53BP1* suggest that most of the analyses of a single tumor biopsy, even at T0, are not adequate to predict the sensitivity or resistance profile to PARPi.

## 4. Discussion

The two main findings of this proof of principle study, technically based on targeted resequencing analysis of ctDNA from HGS-EOC patients, can be briefly summarized as follows: *(i*) at diagnosis, analysis of ctDNA can complement genomic data derived from conventional single tumor biopsy. (*ii*) Longitudinal ctDNA analysis can be used to intercept the emerging genomic architecture of TRm, allowing a more rational second-line treatment.

Early recurrence detection and its rational-based treatment are the two main clinical needs that have hampered till now the improvement of the mortality rate of HGS-EOC. Particularly, the optimization of the second-line therapy has been difficult because of the inability to track tumor evolution and to study the genomic profile of relapse. The work previously carried out by our group [2] and data presented in this study suggest how longitudinal analysis of ctDNA, focusing on both quantitative (i.e., untargeted sWGS analysis) and qualitative features (i.e., targeted resequencing analysis) of ctDNA, can represent a novel strategy to unravel the issues related to the treatment of relapsed disease.

This study is in fact a natural evolution of previous findings in which we used untargeted sWGS to get quantitative information on changes in the amount of ctDNA (i.e., TF) during patients’ follow-up. Data previously reported by our group clearly demonstrated that analysis of SCNA in plasma samples of HGS-EOC patients can be reliably used in the clinic as a tool to intercept the early onset of disease recurrence. By this approach, we have demonstrated that the tumor recurrence (TRm) can be intercepted up to 491 days before conventional biochemical and radiological assays, thus indicating TF as a reliable biomarker to detect the relapse. This temporal gap between TRm and TRc offers a unique opportunity and a temporal window wide enough to plan treatment of relapsed disease with a more rational approach, rather than a measure of the extension of PFI. However, TF is not informative on the possible therapeutic strategies that can be adopted to treat the relapsed relapsed disease.

The detection of new pathogenic mutations through the targeted ctDNA analysis could help identify actionable variants or alterations that could confer sensitivity or resistance to drugs, thus optimizing treatment choice.

In line with current literature, we have previously demonstrated [26] that a single tumor biopsy does not mirror the complex genomic portrait of the disease, and that the genomic profile taken at time of relapse is markedly different (less than 5%) from that of matched primary synchronous lesions naive to chemotherapy. In line with these findings, data reported in this study confirmed that less than 2% of SNVs found in the plasma at T0 are shared by at least one matched synchronous lesion. This means that the genomic portrait of the residual disease that has been exposed to Pt-based chemotherapy is completely different from the one derived from the biopsy withdrawn at time of surgery. Pt-based chemotherapy provides a positive selective pressure on the outgrowth of subclonal variants that are responsible for relapsed disease, thus evidence that the genomic information taken at the diagnosis is not necessarily useful to plan the second line of treatment. In this study, we have demonstrated that—starting from the same sWGS genomic libraries previously analyzed—it is possible to obtain genomic information characterizing the molecular relapse.

Over the last years, the introduction of PARPi in the current clinical practice has changed the therapeutic scenario available for HGS-EOC. Although the majority of patients eligible respond to PARPi treatment, a proportion of them show an intrinsic or acquired resistance. The biological mechanisms related to resistance against PARPi are different and include genomic alterations—e.g., *BRCA1/2* reverse mutations or concomitant pathogenic mutations in *BRCA1* and *TP53BP1* genes—that could exist already at time of diagnosis or that could be acquired under a clonal selective pressure. The analysis of ctDNA both at the time of diagnosis and during patients’ follow-up makes it possible to intercept the presence of resistant clones, thus potentially avoiding unnecessary treatment to patients. As an example, patient 21556, who carries a germline mutation in *BRCA1*, could be potentially eligible for PARPi therapy, but the targeted analysis of cfDNA at the time of diagnosis revealed the presence of two pathogenic variants in the *TP53BP1* gene not detected in the tumor lesions analyzed. The present study shows the potential importance of longitudinal ctDNA analysis to follow dynamic changes in the SNVs profile of selected genes. For example, patient 21564, potentially eligible for PARPi treatment at the time of diagnosis *(BRCA1* c.5182delA mutation), would be unlikely to respond to these drugs in second-line therapy due to the presence of eight distinct *TP53BP1* variants in plasma sample collected and analyzed at TRc. From a clinical point of view, this result could help avoid unnecessary treatment to a patient who has a low probability of response to PARPi exposure.

Finally, from a technical point of view, we have developed a two-step framework of analysis based on a combination of two different NGS-based strategies starting from a single plasma withdrawn. As HGS-EOC is characterized by a marked genomic instability, sWGS has been demonstrated to be the optimal tool to routinely monitor the extension of PFI and provide quantitative information on the early increase in the amount of ctDNA present in the bloodstream, and thus intercept the early molecular footprints of TRm. At that time, it is important to get qualitative information on how a tumor evolves over time after therapeutic positive fitness and which are the main genomic vulnerabilities that will characterize the emerging resistant metachronous lesions that could be exploited for therapeutic intervention. At this step, a high-depth targeted resequencing approach on the same genomic libraries previously generated for sWGS represents the optimal strategy to provide information in a pre-designed and updated panel of genes, the role of which is important for therapeutic intervention. This double-hit strategy is versatile, cheap and compatible with clinical turnaround times required for the management of relapse of HGS-EOC patients.

This proof of principle study faces different limits that should be taken into consideration before its use in clinical practice. First, the small sample size limits the generalizability of our findings. Second, cases have been recruited within a spontaneous study rather than within the frame of a clinical trial in which the rigid clinical protocol guarantees homogenous time points and standard methods for sample collection. Third, we have no information on the kineticity of ctDNA and the internal and external factors that could influence the total amount of ctDNA. Thus, we do not know the minimal amount of ctDNA present in an aliquot of plasma necessary to decisively call rare ctDNA molecules. Fourth, panel selection has been arbitrary, as there are no standard-of-care targetable biomarkers demonstrated to predict at progression, response to second-line treatment. In line with this caveat, the clinical utility of our approach has not been established yet. There is no evidence that treatment planned on the basis of the detection of ctDNA will improve the response of relapsed disease. It is expected that future prospective clinical trials with a large cohort of cases will allow to estimate the sensitivity and accuracy of this approach, as well as its clinical utility.

## 5. Conclusions

In conclusion, data presented in this work are a proof of concept to demonstrate how the use of liquid biopsy could represent a new avenue to overcome the issues of spatial and temporal heterogeneity that has long limited the improvement of the clinical outcome of HGS-EOC patients. 

## Figures and Tables

**Figure 1 cancers-14-03697-f001:**
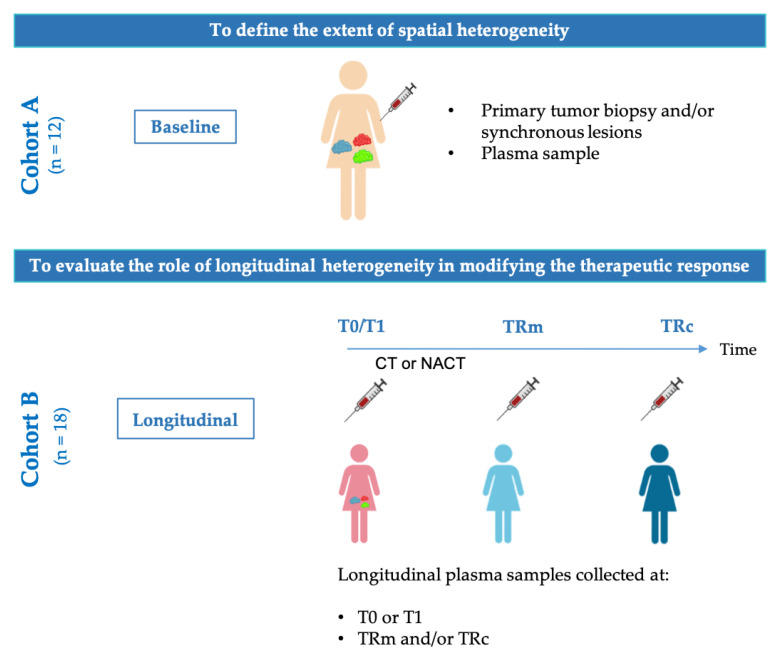
Study overview. The study is mainly subdivided into two different sections focused on defining the extension of spatial (cohort A, *n* = 12) and temporal (cohort B, *n* = 18) heterogeneity in HGS-EOC patients. CT, chemotherapy. NACT, neoadjuvant chemotherapy, T0, time of diagnosis. T1, first time point after diagnosis. TRm, time of molecular recurrence. TRc, time of clinical recurrence.

**Figure 2 cancers-14-03697-f002:**
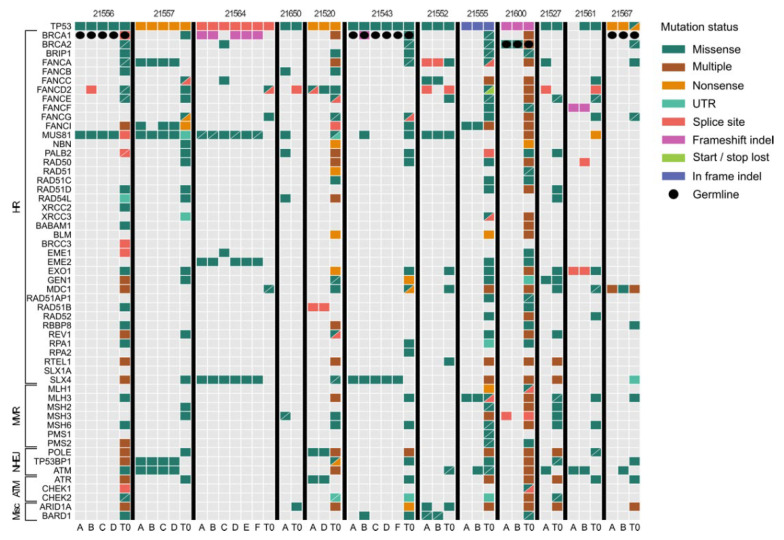
The different genomic profiles of liquid and solid biopsies at the time of diagnosis. For cases enrolled within cohort A (columns), oncoprint depicts all non-synonymous variants called in the coding sequence of the selected 65 genes (horizontal lines). For each patient, A, B, C, D identify each single tumor biopsy (see Appendix A), while T0 refers to the ctDNA analysis. The details of the different variants are reported in the legend on the right side. Gray block means wild type.

**Figure 3 cancers-14-03697-f003:**
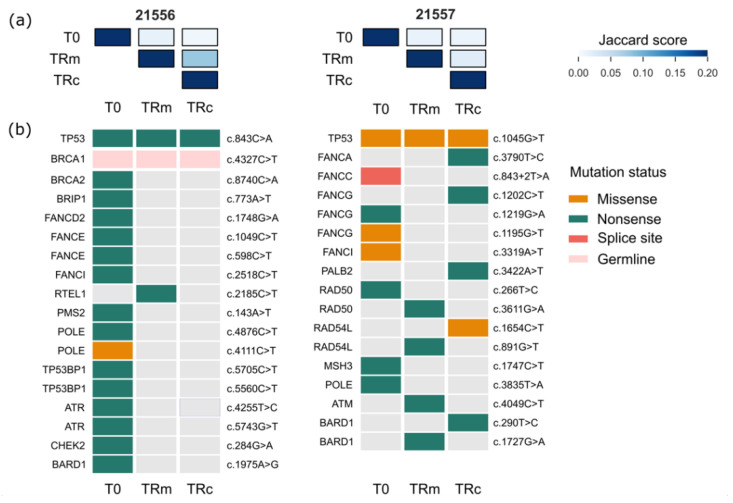
Genomic landscape evolution of SNVs in plasma samples. Jaccard similarity score (**a**) and oncoprints of pathogenic variants (**b**) for two representative cases (namely, 21556 and 21557) are reported. (**a**) The Jaccard similarity score is a numeric value indicating the level of similarity of sample pairs (see Section 2). Rows and columns report the name of the samples as in a square matrix (i.e., T0, TRm and TRc). The *Jaccard score* is reported in a blue color scale on the right side of the figure. The darker the color, the higher the similarity score, as reported in the legend (min = 0%, max = 20%). (**b**) Oncoprints show longitudinal evolution (i.e., T0, TRm and TRc) of pathogenic variants in each plasma sample. Variants are represented with the HGVS notation and referred to the genes reported on the left. The legend on the right depicts the different SNVs identified.

**Figure 4 cancers-14-03697-f004:**
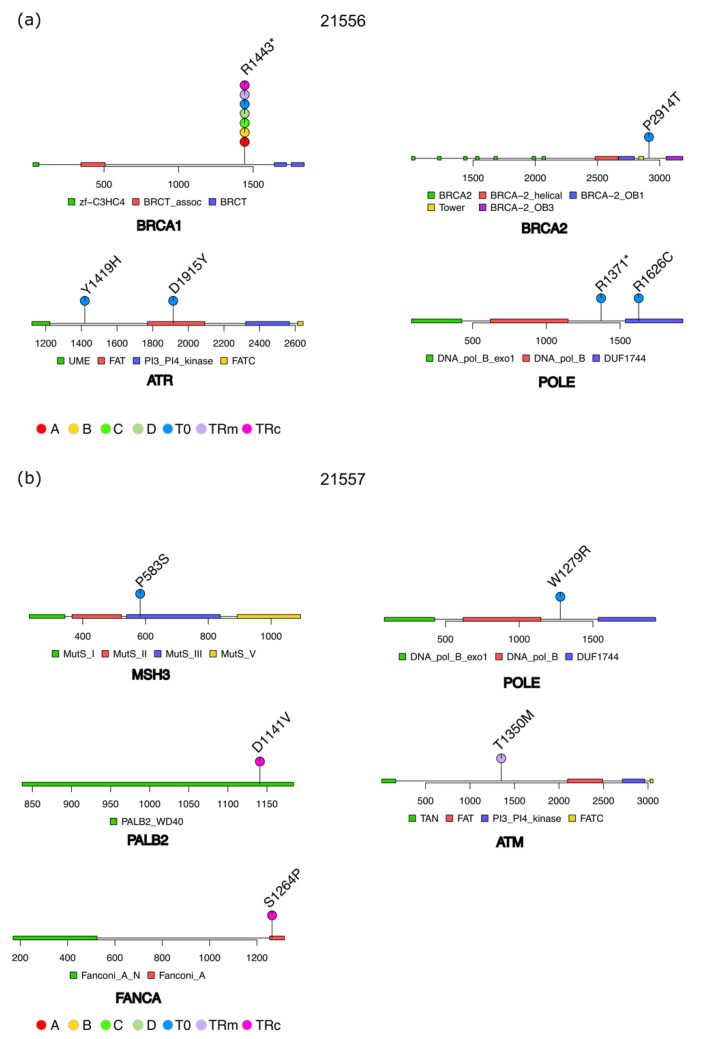
Lolliplots showing actionable variants in two representative patients, namely, 21556 (**a**) and 21557 (**b**). The name of encoded proteins is reported below each graph and the related protein domains, as described in the legend. Actionable variants are reported in correspondence of their amino acid position and named as reported above each dot. Dots represent the presence of an actionable variant in tumor and/or plasma samples, as indicated by the color in the legend.

**Figure 5 cancers-14-03697-f005:**
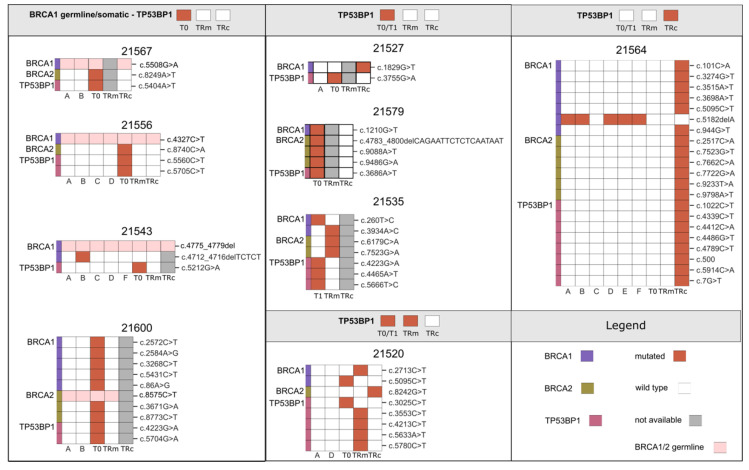
Oncoprints of 13 patients carrying pathogenic mutations in the *TP53BP1* gene. For each oncoprint, mutated genes, e.g., *BRCA1/2* or *TP53BP1*, are reported on the left; codon change is reported on the right; the patient number is reported on the top, and the name of each sample at the bottom. Colors as reported in the legend.

**Table 1 cancers-14-03697-t001:** Clinical, pathologic, and demographic features of cases enrolled in the study ^1^.

Annotations	
Number of patients	18
Median age (years)	64
Age range (years)	48–79
Follow-up time years (range)	3 (1–5)
Histologic type (%)	
Serous	18 (100)
FIGO classification (%)	
III NA	-
III A	-
III B	1 (5)
III C	11 (61)
IV	6 (34)
Pt status (%)	
Sensitive (PFI > 12 mos.)	11(61)
Partially Sensitive (6 mos. < PFI ≤ 12 mos.)	7 (39)
Resistant (1 mo. < PFI ≤ 6 mos.)	-
Refractory (PFI ≤ 1 mo.)	-
NA	-
Chemotherapy (%)	
NACT	7 (39)
CT	11 (61)
Line of chemotherapy (%)	
I	3 (17)
II	7 (38)
III	2 (11)
IV	6 (34)
*BRCA* germline status (%)	
*BRCA1* mut	4 (22)
*BRCA2* mut	1 (6)
*BRCA1/2* wt	9 (50)
NA	4 (22)
Total number of tumor biopsies	33
Total number of plasma samples	43
Number of plasma samples at different time points (%)	
T0	15 (35)
T1	3 (7)
TRm	13 (31)
TRc	11 (25)
TRm = TRc	1 (2)

^1^ The table summarizes the main clinical, pathologic, and demographic features of patients enrolled in this study. Mut, mutant. Wt, wild-type. T0, time of diagnosis. T1, first time available after diagnosis (mean: 12 months after T0). TRm, time of molecular recurrence. TRc, time of clinical recurrence. PFI, Pt-free interval. Mo(s)., month(s). NACT, neoadjuvant chemotherapy. CT, chemotherapy.

## Data Availability

Shallow whole-genome sequences are available under controlled access at the European Genome-Phenome Archive (EGA) with ID EGAS00001004670. Hybrid capture raw sequence data are available at Zenodo (accession pending) under controlled access.

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
