# Peer review of "Targeted Mutational Analysis of Circulating Tumor DNA to Decipher Temporal Heterogeneity of High-Grade Serous Ovarian Cancer"

_cancers, 2022, doi:10.3390/cancers14153697_

Round 1

Reviewer 1 Report

This study analyzed molecular alterations from tissue and liquid biopsies of HGS-EOS patients from two cohorts. Cohort A for spatial and B for temporal heterogeneity. Overall, the data is good, the presentation can be simplified for readers to follow with more easy-to-read figures.  

The Graphical Abstract can be made clear for the reader. Is this patient-level figure or aggregate data? The reader may be a bit confused looking at the mutations described in a longitudinal fashion.

  Abstract- The results need to be made more clear and detailed in the abstract highlighting the value/main conclusions of the study.   Figure 2 is very hard to read. My suggestion is to just show the overall prevalence for each gene separately and then do an oncoplot like the one you have for the interesting genes or top prevalent genes only. This will simplify the figure further. Also, concordance between tissue and liquid can vary depending on how close the tissue biopsy and blood draws were obtained. Was this information for each patient available? Figure 2b- I am unable to understand this figure or the take-home message here.   Figure 3- The Jaccard scoring is not clear in figure 3. Clarifying what each box represents in figure 3a and detailing the main conclusions would be helpful to readers. Figure 3b is clear. Why were these two representative cases selected?   Figure 4- again please provide a rationale for selecting these patients. Were these the only pts with actionable alterations? Please provide the overall prevalence of actionable alterations.   Results Section 3.2- The selection of 65 genes was based on main pathways that were reported, were studies in the literature that did comprehensive genomic sequencing used as a rationale as well?   Discussion- consider adding a section detailing the limitations of the study for the readers. One could include a small sample size as one factor. Secondly, the time of blood draw is an important factor to consider while determining tumor DNA fraction. Was this accounted for? If blood was drawn during chemo then high shedding can be expected.     Conclusions - More clarity can be helpful for readers. Referring to the first paragraph under discussion, how many cases were identified where ctDNA analysis revealed more alterations? second point- how many patients in aggregate was longitudinal monitoring useful? I could not find this in the text.

Reviewer 2 Report

The manuscript entitled “Targeted mutational analysis of circulating tumor DNA to decipher temporal heterogeneity of high-grade serous ovarian cancer” analyzed the circulating tumor DNA from patients diagnosed with high-grade serous ovarian cancer using a targeted resequencing approach. This is an interesting and well-written work.

Major points:

- In the Material and Methods section is not described how tumor DNA was extracted from primary tumors and synchronous lesions. This information is important to data interpretations. If DNA was extracted from a single spot instead of multiple spots from different locations within the tumor, the heterogeneity found will be lower and may increase the discordance between tumor samples and ctDNA samples. If only one single tumor spot were used to extract tumor DNA, this fact must be addressed in the Discussion section.

- The study limitations should be included in the Discussion section.

Minor points:

- In the abstract, TP53 should be italicized (page 1, line 36).

- In targeted resequencing analysis, were studied 66 (page 4, line 114) or 65 genes (page 7, line 204)?

- In Table 1, the text after table legend should be placed as a footnote at the end of the table.

- In Figure 1 legend, NACT should be written out in full.

- In line 295 (page 10), “avariant” should be replaced by “a variant”.

Round 2

Reviewer 1 Report

All revisions are acceptable to me

Reviewer 2 Report

The Authors adequately addressed the raised comments. In Abstract, TP53BP1 (lines 46 and 49) and BRCA1/2 (lines 43 and 48) genes should be italicized. I have no further comments.